# ENERGY-BASED PREDICTIVE REPRESENTATIONS FOR PARTIALLY OBSERVED REINFORCEMENT LEARNING

## ABSTRACT

In real world applications, it is usually necessary for a reinforcement learning algorithm to handle partial observability, which is not captured by a Markov decision process (MDP). Although partially observable Markov decision processes (POMDPs) have been precisely motivated by this requirement, they raise significant computational and statistical hardness challenges in learning and planning. In this work, we introduce the *Energy-based Predictive Representation (EPR)* to support a unified approach to practical reinforcement learning algorithm design for both the MDP and POMDP settings, which enables *learning, exploration, and planning* to be handled in a coherent way. The proposed framework relies on a powerful neural energy-based model to extract a sufficient representation, from which $Q$-functions can be efficiently approximated. With such a representation, confidence can be efficiently computed to allow optimism/pessimism in the face of uncertainty to be efficiently implemented in planning, enabling effective management of the exploration versus exploitation tradeoff. An experimental investigation shows that the proposed algorithm can surpass state-of-the-art performance in both MDP and POMDP settings in comparison to existing baselines.

## 1 INTRODUCTION

Reinforcement learning (RL) based on Markov Decision Processes (MDPs) has proved to be extremely effective in several real world decision-making problems (Levine et al., 2016; Jiang et al., 2021). However, the success of most RL algorithms (Ren et al., 2022b; Zhang et al., 2022) relies heavily on the assumption that the environment state is fully observable to the agent. In practice, such an assumption can be easily violated in the presence of observational noise. To address this issue, Partially Observable Markov Decision Processes (POMDPs) (Åström, 1965) have been proposed for capturing the inherent uncertainty about the state arising from partial observations.

However, the flexibility of POMDPs creates significant statistical and computational hardness in terms of planning, exploration and learning. In particular, **i)**, partial observability induces a non-Markovian dependence over the *entire history*; and **ii)**, the expanded spaces of observation sequences or state space distributions incur significant representation challenges. In fact, due to the full history dependence, it has been proved that the planning for even finite-horizon tabular POMDPs is NP-hard without additional structural assumptions (Papadimitriou & Tsitsiklis, 1987; Madani et al., 1998), and the sample complexity for learning POMDPs can be exponential with respect to the horizon (Jin et al., 2020a). These complexities only become more demanding in continuous state spaces and real-world scenarios.

On the other hand, despite the theoretical hardness, the widely used sliding window policy parameterization has demonstrated impressive empirical performance (Mnih et al., 2013; Berner et al., 2019), indicating that there is sufficient structure in real-world POMDPs that can be exploited to bypass the aforementioned complexities. Recently, observable POMDPs with invertible emissions have been investigated to justify the sliding window heuristic in tabular cases (Azizzadenesheli et al., 2016; Guo et al., 2016; Jin et al., 2020a; Golowich et al., 2022a), which has been further extended with function approximation for large and continuous state POMDPs (Wang et al., 2022; Uehara et al., 2022). Although these algorithms can exploit particular structure efficiently in terms of the sample complexity, they rely on unrealistic computation oracles, and are thus not applicable in practice. In this paper, we consider the following natural question:

*How can one design **efficient** and **practical** algorithms for **structured** POMDPs?*

In particular, we would like to exploit special structures that allows approximation to bypass inherent worst-case difficulties. By "efficient" we mean considering **learning**, **planning** and **exploration** in a unified manner that can balance errors in each component and reduce unnecessary computation, while by "practical" we mean the algorithm retains sufficient flexibility and can be easily implemented and deployed in real-world scenarios.

There have been many attempts to address this question. The most straightforward idea is to extend model-free RL methods, including policy gradient and $Q$-learning, with a memory-limited parametrization, *e.g.*, recurrent neural networks (Wierstra et al., 2007; Hausknecht & Stone, 2015; Zhu et al., 2017). Alternatively, in model-based RL (Kaelbling et al., 1998), an approximation of the latent dynamics can be estimated and a posterior over latent states (*i.e.*, beliefs) maintained, in principle allowing an optimal policy to be extracted via dynamic programming upon beliefs. Following this idea, Deisenroth & Peters (2012) and Igl et al. (2018); Gregor et al. (2019); Zhang et al. (2019); Lee et al. (2020) consider Gaussian process or deep model parametrizations, respectively. Such methods are designed based on implicit assumptions about structure through the parameterization choices of the models. However, these approaches suffer from sub-optimal performance due to several compounding factors: **i)**, approximation error from inaccurate parametrizations of the learnable components (policy, value function, model, belief), **ii)**, a sub-optimal policy induced by approximated planning (through policy gradient or dynamic progamming), and **iii)**, the neglect of exploration when interacting with the environment.

As an alternative, spectral representation approaches provide an alternative strategy based on extracting a sufficient representation that can support learning, planning and exploration. In this vein Azizzadenesheli et al. (2016) investigate spectral methods (Anandkumar et al., 2014) for latent variable model estimation in POMDPs, but only consider tabular scenarios with finite state and action cases. Predictive State Representations (PSR) (Littman & Sutton, 2001; Boots et al., 2011) also leverage spectral decomposition, but instead of recovering an underlying latent variable model, they learn an equivalent sufficient representation of belief. These methods have been extended to real-world settings with continuous observations and actions by exploiting kernel embeddings (Boots et al., 2013) or deep models (Downey et al., 2017; Venkatraman et al., 2017; Guo et al., 2018). However, efficient exploration and tractable planning with spectral representations has yet to be thoroughly developed (Zhan et al., 2022).

In this paper, we propose *Energy-based Predictive Representation (EPR)* to support efficient and tractable learning, planning, and exploration in POMDPs (and MDPs), as a solution to the aforementioned question. More specifically:

- We propose a flexible nonlinear energy-based model for induced belief-state MDPs *without explicit parameterization of beliefs*, providing a principled *linear sufficient* representation for the state-action value function.
- We reveal the connection between EPR and PSR, while also illustrating the differences, to demonstrate the modeling ability of the proposed EPR.
- We provide *computationally-tractable* learning and planning algorithms for EPR that implement the principles of optimism and pessimism in the face of uncertainty for online and offline RL, balancing exploration and exploitation.
- We conduct a comprehensive comparison to existing state-of-the-art RL algorithms in both MDP and POMDP benchmarks, demonstrating superior empirical performance of the proposed EPR.

## 2 PRELIMINARIES

In this section, we briefly introduce POMDPs and their degenerate case of MDPs, identifying the special structures that will be used to derive the proposed representation learning method.

**Partially Observable Markov Decision Processes.** Formally, we define a partially observable Markov decision process (POMDP) as a tuple $\mathcal{P} = (\mathcal{S}, \mathcal{A}, \mathcal{O}, r, H, \mu, P, O)$, where $H$ is a positive integer denoting the length of horizon; $\mu$ is the initial distribution of state, $r : \mathcal{S} \times \mathcal{A} \to [0, 1]$, the reward function, and $\mathcal{S}, \mathcal{A}, \mathcal{O}$ denote the state, action and observation space, respectively. $P(\cdot|s, a) : \mathcal{S} \times \mathcal{A} \to \Delta(\mathcal{S})$ is the transition kernel, capturing the dynamics between states, and $O(\cdot|s) : \mathcal{S} \to \Delta(\mathcal{O})$ is the emission kernel, where $\Delta(\cdot)$ denotes the set of probability measures over the support.

Initially, given a state $s_1 \sim \mu(s)$ as a starting point, at each step $h \in [1, H]$, the agent takes an action $a \in \mathcal{A}$, a new state $s_{h+1}$ is generated $s_{h+1} \sim P(\cdot|s_h, a_h)$, and the agent observes $o_{h+1} \sim O(\cdot|s_{h+1})$

and reward $r(s_{h+1}, a_{h+1})$. Due to partial observability, the dependence between observations is non-Markovian, hence, we define a policy $\pi = \{\pi_t\}$ where $\pi_t : \mathcal{O} \times (\mathcal{A} \times \mathcal{O})^t \to \Delta(\mathcal{A})$ to depend on the whole history, *i.e.*, $x_t = \{o_0, \{a_i, o_{i+1}\}_{i=0}^{t-1}\}$. The corresponding value for policy $\pi$ can be defined as $V^\pi = \mathbb{E}_\pi \left[ \sum_{h=1}^H r(s_h, a_h) \right]$, and the objective is to find the optimal policy $\pi^* = \arg\max_\pi V^\pi$.

Markov decision processes (MDPs) are a degenerate case of POMDPs, where $\mathcal{S} = \mathcal{O}$ and $O(o|s) = \delta(o = s)$, and can be specified as $\mathcal{M} = (\mathcal{S}, \mathcal{A}, r, H, \mu, P)$. One can also convert a POMDP to an MDP by treating the whole history $x_t = \{o_0, \{a_i, o_{i+1}\}_{i=0}^{t-1}\}$ as the state. Specifically, following (Kaelbling et al., 1998), we define the belief $b : \mathcal{O} \times (\mathcal{A} \times \mathcal{O})^t \to \Delta(\mathcal{S}), \forall t \in \mathbb{N}^+$, which can be recursively defined as: $b(s_1|o_1) = P(s_1|o_1)$, and

$$b(s_{t+1}|x_{t+1}) = \frac{b(s_t|x_t)P(s_{t+1}|s_t,a_t)O(o_{t+1}|s_{t+1})}{\int b(s_t|x_t)P(s_{t+1}|s_t,a_t)O(o_{t+1}|s_{t+1})\,\mathrm{d}s_t\,\mathrm{d}o_{t+1}}. \tag{1}$$

Each entry of the belief state describes the probability of the underlying state given the past history. Furthermore, with a slight abuse of notation, we use $b_t$ to denote the belief state at step $t$. Then, one can construct the equivalent belief MDP $\mathcal{M}_b = (\mathcal{X}, \mathcal{A}, R_h, H, \mu_b, T_b)$ with $\mathcal{X}$ denoting the set of possible histories, and

$$\mu_b := \int b(s|o_1)\mu(o_1)do_1, \quad R_t(b,a) = \int b_t(s_h)r(s_t,a)ds_h \tag{2}$$

$$T_b(b_{t+1}|b_t,a_t) := \int_{\mathcal{O}} \mathbf{1}_{b_{t+1}=b(x_{t+1})} P(o_{t+1}|b_t,a_t)\,\mathrm{d}o_{t+1}. \tag{3}$$

Therefore, the corresponding value function $V_h^\pi(b_h)$ and $Q_h^\pi(b_h, a_h)$ for the belief MDP given a policy $\pi$ can be defined as:

$$V_h^\pi(b_h) = \mathbb{E}\left[\sum_{t=h}^H R_t(b_t,a_t)|x_h\right], \quad Q_h^\pi(b_h,a_h) = \mathbb{E}\left[\sum_{t=h}^H R_t(b_t,a_t)|b_h,a_h\right].$$

Following the MDP perspective, we also have the Bellman recursive equation:

$$V_h^\pi(b_h) = \mathbb{E}_\pi[Q_h^\pi(b_h,a_h)], \quad Q_h^\pi(b_h,a_h) = R_h(b_h,a_h) + \mathbb{E}_{T_b}\left[V_{h+1}^\pi(b_{h+1})\right]. \tag{4}$$

One can still apply a dynamic programming style approach to solve POMDPs according to (4), however since the belief depends on the entire history the number of possible beliefs can still be infinite even the number of states is finite. To combat with these essential difficulties, we will leverage two particular structures, observability and linearity, as introduced below.

**Observability in POMDPs.** It has been shown (Even-Dar et al., 2007; Golowich et al., 2022b; Uehara et al., 2022) that for POMDPs with an observability assumption, one can safely relax the history dependence with a short window, bypassing the exponential sample and planning complexity w.r.t. horizon length (Golowich et al., 2022a;b). Specifically, the observability property for POMDPs is defined as follows.

**Assumption 1** ((Even-Dar et al., 2007; Golowich et al., 2022b)). *The POMDP with emission model $O$ satisfies $\gamma$-observability if for arbitrary beliefs $b$ and $b'$ over states, $\|\langle O, b\rangle - \langle O, b'\rangle\|_1 \geq \gamma \|b - b'\|_1$, where $\langle O, b\rangle := \int O(o|s)b(s)ds$.*

A key consequence of observability is that, the belief can be well approximated with a short history window (Golowich et al., 2022b), and one can construct an approximate MDP based on a finite belief history, which eliminates the exponential complexity induced by full history dependence. Specifically, we denote $L$ as the length of the window. Then, defining $x_t^L = \left\{o_{t-L}, \{a_i, o_{i+1}\}_{i=t-L}^t\right\} \in \mathcal{X}^L$, the approximated beliefs $b^L$ follow the same recursive definition as (1) but with only finite history $x_t^L$ starting from the uniform belief. This immediately induces an approximate MDP $\mathcal{M}_b^L = \left(\mathcal{X}^L, \mathcal{A}, R_h^L, H, \mu_b, T_b^L\right)$ according to (3) with $b^L$, instead of $b$. Theorem 2.1 in Golowich et al. (2022a) proves that the approximation error of the finite-memory belief MDP is small for observable POMDPs. Hence, with slight abuse of notation, we still use $b$ to represent $b^L$ throughout the paper.

**Linearity in MDPs.** To handle the complexity induced by large state spaces, linear/low-rank structures have been introduced in MDPs (Jin et al., 2020b; Agarwal et al., 2020) for effective function approximation, which leverages spectral factorization of the transition dynamics and reward:

$$P(s'|s,a) = \langle \phi(s,a), \mu(s')\rangle, \quad r(s,a) = \langle \phi(s,a), \theta\rangle, \tag{5}$$

where $\phi : \mathcal{S} \times \mathcal{A} \to \mathcal{H}, \mu : \mathcal{S} \to \mathcal{H}$ are two feature maps to a Hilbert space $\mathcal{H}$. Under such an assumption, we can represent the state-action value function $Q^\pi$ for an arbitrary policy $\pi$ by:

$$Q^\pi(s,a) = r(s,a) + \gamma \int V^\pi(s')P(s'|s,a)ds' = \left\langle \phi(s,a), \theta + \gamma \int V^\pi(s')\mu(s')ds'\right\rangle,$$

which implies that instead of a complicated function space defined on the raw state space, one can design a computationally efficient planning and sample efficient exploration algorithm in the space linearly spanned by $\phi$. In fact, from the correspondence between policy and $Q$-function as discussed in (Ren et al., 2022a), $\phi$ can be understood as representing primitives for skill set construction. Efficient and practical algorithms have been designed for exploiting linearity in MDPs (Zhang et al., 2022; Qiu et al., 2022), which inspires us to exploit similar properties in POMDPs.

**Energy-based Models.** Energy-based Models are one of the most flexible models to represent the conditional probability measure. It takes the form of $p(y|x) = \exp(-f(x,y))/Z(x)$ where $f(x,y)$, which can be parametrized by deep models, is the energy of $(x, y)$ and $Z(x)$ is a partition function that only depends on $x$ to guarantee $p(y|x)$ is a valid probability measure. When $y$ is discrete, we have that $p(y|x) = \exp(-f(x,y))/\sum_y \exp(-f(x,y))$, which corresponds to the standard softmax probability where $-f(x,y)$ is the softmax logits. We refer the interested readers to Song & Kingma (2021) for the training methods of energy-based models.

## 3 ENERGY-BASED PREDICTIVE REPRESENTATION

We propose *Energy-based Predictive Representation (EPR)*, which introduces linearity into finite-history approximated POMDPs, allowing the complexity induced by large state spaces and long histories to be overcome, yielding improved efficiency for learning, planning and exploration. We emphasize that the proposed method is also applicable to MDPs.

The approach builds upon recent progress in large-state MDPs (Zhang et al., 2022; Qiu et al., 2022) that leverages linear structure in the dynamics, $P(s'|s,a) = \langle \phi(s,a), \mu(s') \rangle$, to obtain an efficient and practical framework for learning, planning and exploration. Recall the construction of a finite-memory belief MDP to approximate a POMDP discussed in Section 2, which avoids full history dependence. For such a constructed belief MDP, a natural idea is to apply linear MDP algorithms, *i.e.*, extracting the linear decomposition for $T_b^L(b'|b,a) = \langle \phi(b,a), \mu(b') \rangle$, to handle the hardnesses of POMDPs mentioned in Section 1. However, there are several difficulties in such a straightforward extension:

  **i**, the set of beliefs is proportional to the number of states, which could be infinite;
  **ii**, the factorization of the transition dynamics (3) in the belief MDP is difficult.

These difficulties hinder the extension of linear MDPs to observable POMDPs. However, note that we never explicitly require the beliefs and their dynamics, but only the representation $\phi(b, a)$. As beliefs are functions over finite-window histories, the representation can also be rewritten as $\phi(x_t, a_t)$, which suggests that one might bypass the inherent difficulties by a *reprameterization trick*. Consider the energy-based parametrization (Arbel et al., 2020) for $P(o_{t+1}|b(x_t), a_t)$ where $b(x_t)$ is the belief for history $x_t$:

$$P(o_{t+1}|b(x_t), a_t) = p(o_{t+1}) \exp\left(f(x_t, a_t)^\top \left(g(o_{t+1}) + \lambda f(x_t, a_t)\right)\right), \qquad (6)$$

$$\mathbb{E}_{o_{t+1}}\left[\exp\left(f(x_t, a_t)^\top \left(g(o_{t+1}) + \lambda f(x_t, a_t)\right)\right)\right] = 1, \forall (x_t, a_t) \in \mathcal{X}^L, \qquad (7)$$

where $\lambda$ is a scalar, $p(o)$ is a fixed distribution and the normalization condition enforces that the energy-based model $P(o_{t+1}|b(x_t), a_t)$ is a valid distribution. We avoid any explicit parametrization and computation of beliefs $b$, while preserving dependence through $f$ and $g$, which will be learned jointly. Compared to standard parametrizations, we do not need to specify unnecessary model parameters for the transition dynamics $P$ and emmission $O$, and bypass any learning and approximation of beliefs that induce compounding errors. As a special case, we note that the observable Linear-Quadratic Gaussian (LQG) actually follows (6) with a specific $\lambda$ and $p(o)$. See Appendix C for details.

Meanwhile, this approach also provides a linear factorization of $T(b_{t+1}|b_t, a_t)$ almost for free. By viewing the proposed parameterization (6) as a kernel and following the random Fourier feature trick (Rahimi & Recht, 2007; Choromanski et al., 2020; Ren et al., 2022b), one can write

$$P(o_{t+1}|b_t, a_t) = \mathbb{E}_\omega\left[\phi_\omega(x_t, a_t)\psi_\omega(o_{t+1})\right], \qquad (8)$$

where $\omega_i \sim \mathcal{N}(0, I_d)$ and

$$\phi(x_t, a_t) = \left[\exp\left(\left(\lambda - \frac{1}{2}\right)\|f(x_t, a_t)\|^2 + \omega_i^\top f(x_t, a_t)\right)\right]_{i=1}^d, \qquad (9)$$

$$\psi(x_{t+1}) = \left[p(o_{t+1}) \exp\left(\omega_i^\top g(o_{t+1}) - \frac{\|g(o_{t+1})\|^2}{2}\right)\right]_{i=1}^d, \qquad (10)$$

which can be derived from the softmax random feature from Choromanski et al. (2020). We also provide a derivation in Appendix B. Substituting (8) into (3) yields the factorization of $T_b$ as

$$T_b\left(b_{t+1}|b_t, a_t\right) = \int_{\mathcal{O}} \mathbf{1}_{b(\cdot|x_{t+1})=b(\cdot|x_t, a_t, o_{t+1})} \mathbb{E}_\omega\left[\phi_\omega(x_t, a_t)\psi_\omega(o_{t+1})\right] \mathrm{d}o_{t+1} \tag{11}$$

$$= \mathbb{E}_\omega\left[\phi_\omega(x_t, a_t)\mu(b_{t+1})\right], \tag{12}$$

where $\mu(b_{t+1}) := \int_{\mathcal{O}} \mathbf{1}_{b(\cdot|x_{t+1})=b(\cdot|x_t, a_t, o_{t+1})}\psi_\omega(o_{t+1}) \, \mathrm{d}o_{t+1}$. With this formula, we obtain a valid linear representation $\phi(b_t, a_t)$ as an Energy-based Predictive Representation (EPR) for a belief MDP without any explicit beliefs.

To learn the EPR given data $\mathcal{D} := \{o_{t-1}, a_t, r_t\}_{t=1}^H$, we exploit maximum likelihood estimation (MLE) of (6),

$$\min_{f,g} \quad -\widehat{\mathbb{E}}_{\mathcal{D}}\left[f(x_t, a_t)^\top\left(g(o_{t+1}) + \lambda f(x_t, a_t)\right)\right], \tag{13}$$

$$\text{s.t.,} \quad \mathbb{E}_{p(o_{t+1})}\left[\exp\left(f(x_t, a_t)^\top\left(g(o_{t+1}) + \lambda f(x_t, a_t)\right)\right)\right] = 1, \forall\,(x_t, a_t) \in \mathcal{X}^L. \tag{14}$$

To ensure the constraints, we add a regularization term

$$\left(\log\left(\mathbb{E}_o\left[\exp(f(x_t, a_t)^\top\left(g(o) + \lambda f(x_t, a_t)\right))\right]\right)\right)^2 \tag{15}$$

$$\approx \left(\log\left(\frac{1}{m}\sum_{i=1}^m \exp(f(x_t, a_t)^\top\left(g(o_i) + \lambda f(x_t, a_t)\right))\right)\right)^2, \tag{16}$$

with $o_i \sim p$. The objective will be

$$\min_{f,g} \ \widehat{\mathbb{E}}_{\mathcal{D}}\bigg[-f(x_t, a_t)^\top\left(g(o_{t+1}) + \lambda f(x_t, a_t)\right)$$

$$+ \alpha\left(\log\left(\frac{1}{m}\sum_{i=1}^m \exp(f(x_t, a_t)^\top\left(g(o_i) + \lambda f(x_t, a_t)\right))\right)\right)^2\bigg]. \tag{17}$$

In practice, we can further simplify the objective by normalizing the $\tilde{f}(x_t, a_t) = \frac{f(x_t, a_t)}{\|f(x_t, a_t)\|_2}$, obtaining the final objective

$$\min_{\tilde{f},g} \ \widehat{\mathbb{E}}_{\mathcal{D}}\left[-\tilde{f}(x_t, a_t)^\top g(o_{t+1}) + \lambda + \alpha\left(\log\left(\sum_{i=1}^m \exp(\tilde{f}(x_t, a_t)^\top g(o_i) + \lambda)\right)\right)^2\right], \tag{18}$$

which reduces to a contrastive loss that can be optimized by stochastic gradient descent with a deep network parameterization of $\tilde{f}$ and $g$. We obtain negative samples $\{o_i\}_{i=1}^m \sim p(o)$ from a mixture of replay buffer and collected trajectories.

Before we introduce an exploration-exploitation mechanism with EPR in Section 3.1, we first discuss the relationship between the proposed EPR, predictive state representations (PSR) (Littman & Sutton, 2001; Singh et al., 2004), and spectral dynamics embedding (SPEDE) (Ren et al., 2022b).

**Connection to PSR (Littman & Sutton, 2001; Singh et al., 2004):** The predictive state representation (PSR) was also proposed for bypassing belief calculation by factorizing a variant of the transition dynamics operator. Specifically, given the history $(x_t, a_t)$, the probability for observing a test, *i.e.*, the finite sequence of events $y = (a_{t+1}, o_{t+1}, \cdots, a_{t+k}, o_{t+k})$ with $k \in \mathbb{N}$, is $p(y|x) := p(o_{t+1}^{t+k}|x_t, a_t^{t+k})$. For time step $t$, one can construct a set of core tests $U = \{u_i, \dots, u_k\}$ as sufficient statistics for history $x_t$, such that for any test $\tau$, $p(\tau|x_t) = \langle p(U|x_t), w_\tau \rangle$ for some weights $w_\tau, \in \mathbb{R}^{|U_t|}$. The forward dynamics can be represented in a PSR by Bayes' rule: $p(\tau|x_t, a_t, o_{t+1}) = \frac{w_{(\tau, a_t, o_{t+1})}^\top p(U|x_t)}{w_{(a_t, o_{t+1})}^\top p(U|x_t)}$, which implies that a PSR updates with new observations and actions by repeating a calculation for each $u_i \in U$. Although originally defined for tabular cases, PSRs have been extended to continuous observations by introducing kernels (Boots et al., 2013) or neural networks (Guo et al., 2018; Downey et al., 2017; Hefny et al., 2018).

Obviously, the proposed EPR shares similarities with PSR. Both factorize conditional distributions defined by the dynamics. However, these representations are designed for different purposes, and thus, with different usages and updates. Concretely, EPR is proposed for seeking a linear space that can represent the $Q$-function. The representation is designed to preserve linearity with successive observations without the need for Bayesian updates, which induce extra nonlinearity in PSRs. This

---

**Algorithm 1** Energy-based Predictive Representation

1: **Input:** History Embedding $f(x, a)$, Observation Embedding $g(o)$, Random Feature $\{\omega_i\}_{i=1}^n$ where $\omega_i \sim \mathcal{N}(0, I_d)$, Initial Random Policy $\pi_0$, Initial Dataset $\mathcal{D} = \emptyset$ for online setting.
2: **for** Episode $i = 1, \cdots, K$ **do**
3:     Collect data $\{(x_{i,j}, a_{i,j}, o_{i,j}, r_{i,j})\}_{j=1}^H$ with $\tilde{\pi}_i = \xi\pi_i + (1 - \xi)\pi_0$, and add the data to $\mathcal{D}$.
4:     Optimize $f$ and $g$ with (18) using the data from $\mathcal{D}$.
5:     Obtain the representation $\phi(x_t, a_t)$ via (9) using $\{\omega_i\}_{i=1}^n$.
6:     Add the bonus (19) to the reward and obtain the optimal policy $\pi_{i+1}$ with the $Q(x_t, a_t)$ parameterize as $\phi(x_t, a_t)$ and optimize via FQI.
7:     *(Optional) Extract policy by soft-AC from learned Q.*
8: **end for**
9: **Return** $\pi_K$.

---

linear property leads to efficient exploration and planning in EPR; while an efficient exploration and planning algorithm has not yet been discussed for PSR.

**Connection to SPEDE (Ren et al., 2022b):** Linear random features have been proposed for solving planning in MDPs with nonlinear dynamics in (Ren et al., 2022b), where the transition operator is defined as $T(s'|s, a) \propto \exp\left(-\|s' - f(s, a)\|_2^2/(2\sigma^2)\right)$, corresponding to dynamics $s' = f(s, a) + \epsilon$ with Gaussian noise $\epsilon \sim \mathcal{N}(0, \sigma)$. In addition to the generalization of EPR for POMDPs, even in an MDP, EPR considers a general energy-based model, $T(s'|s, a) \propto p(s') \exp\left(f(s, a)^\top (g(s') + \lambda f(s, a))\right)$ for the dynamics, which is far more flexible than the Gaussian perturbation model considered in SPEDE.

3.1 ONLINE EXPLORATION AND OFFLINE POLICY OPTIMIZATION WITH EPR

With an EPR $\phi(x_t, a_t)$ learned for a POMDP, we can represent the $Q$-function linearly for the approximated belief MDP, and thus, achieve computationally efficient planning, while calculating confidence bounds for implementing the optimism/pessimism in the face of uncertainty.

**Exploration and Exploitation with Elliptical Confidence Bound.** Given the learned representation $\phi(x_t, a_t)$, the confidence bounds can be calculated efficiently, which allows efficient implementation of optimism/pessimism in the face of uncertainty via upper/lower confidence bound (Abbasi-Yadkori et al., 2011; Jin et al., 2020b; Uehara et al., 2021). This is achieved simply by adding an additional elliptical bonus to the $R(x, a)$. Specifically, given the dataset we collect $\mathcal{D} = \{(x_i^L, a_i, R_i, o_{i+1})\}_{i=1}^n$, and calculate the confidence bound as the bonus,

$$b(x_t, a_t) = \sqrt{\phi(x_t, a_t)\Sigma_n^{-1}\phi(x_t, a_t)} \tag{19}$$

where $\lambda$ is a pre-specified hyperparameter, and $\Sigma_n = \sum_{i=1}^n \lambda I + \phi(x_i^L, a_i)\phi(x_i^L, a_i)^\top$. One can then implement UCB/LCB by adding/subtracting the bonus to $R(x_t, a_t)$, and performing planning on the modified reward function.

**Planning with Obtained Representation.** Planning can be conducted by Bellman recursion within the linear space spanned by $\phi(x_t^L, a_t)$ without a bonus. However, with an additional bonus term, the $Q^\pi$ no longer lies in the linear space of $\phi$, since

$$Q^\pi\left(x_t^L, a_t\right) = R(x_t^L, a_t) + b(x_t^L, a_t) + \mathbb{E}_{T_b^L \pi}\left[Q^\pi\left(x_{t+1}, a_{t+1}\right)\right].$$

As discussed in (Zhang et al., 2022), one can augment the feature space $\psi(x, a) := \{\phi(x, a), b(x, a)\}$ to ensure the $Q$-functions can be linearly represented but with an extra $\mathcal{O}(d^2)$ memory cost. In practice, we perform fitted $Q$ iteration with a nonlinear component extending the linear parameterization, *i.e.*, $Q(x, a) = \{w_1, w_2\}^\top [\phi(x, a), \sigma(w_3^\top \phi(x, a))]$.

We provide an outline of our implementation of UCB in Algorithm 1. LCB for pessimistic offline RL is similar but using a pre-collected dataset $\hat{\mathcal{D}}$ without data collection iteration in Step 2, and with the bonus subtracted in Step 6. Our algorithm follows the standard interaction paradigm between the agent and the environment, where for each episode, the agent executes the policy and logs the data to the dataset. Then we perform representation learning and optimistic planning with the $Q$ function parameterized upon the learned representation. Finally, we also extract a policy from the learned $Q$ by soft actor-critic (Haarnoja et al., 2018).

## 4 RELATED WORK

**Partial Observability in Reinforcement Learning.** Despite the essential hardness of POMDPs in terms of learning, planning and exploration (Papadimitriou & Tsitsiklis, 1987; Madani et al., 1998;

Vlassis et al., 2012; Jin et al., 2020b), the study of reinforcement learning with partial observations, from both theoretical and empirical aspects, is still attractive due to its practical importance.

Algorithmically, model-based/-free algorithms have been extended to POMDPs, explicitly or implicitly exploiting structure. Model-based RL algorithms parameterize and learn latent dynamics with an emission model explicitly, and planning through the simulation upon the learned models. A variety of deep models have been proposed recently for better modeling (Watter et al., 2015; Karl et al., 2016; Igl et al., 2018; Zhang et al., 2019; Lee et al., 2020; Hafner et al., 2019a;b; 2020). Although deep models indeed provide better approximation ability, they also bring new challenges in terms of planning and exploration, which has not been fully handled. On the other hand, model-free RL algorithms have been extended for POMDPs by learning history dependent value functions and/or policies, through temporal-difference algorithms or policy gradients. For example, deep $Q$-learning (Mnih et al., 2013) concatenates 4 consecutive frames as the input of a deep neural $Q$-net, which is then improved by recurrent neural networks for longer windows (Bakker, 2001; Hausknecht & Stone, 2015; Zhu et al., 2017). Recurrent neural networks have also been exploited for history dependent policies (Schmidhuber, 1990; Bakker, 2001; Wierstra et al., 2007; Heess et al., 2015) in policy gradient algorithms as well as actor-critic approaches (Ni et al., 2021; Meng et al., 2021). Model-free RL for POMDPs bypasses the planning complexity of model-based RL algorithms. However, the difficulty in exploration remains, which leads to suboptimal performance in practice. By contrast, the proposed EPR not only can be efficiently learned, but is also equipped with simple yet principled planning and exploration methods, which has not been previously achieved.

**Representation Learning for RL.** Successful vision-based representation learning methods have been extended to RL for extracting compact and invariant *state-only* information from raw-pixels, *e.g.*, (Laskin et al., 2020a;b; Kostrikov et al., 2020). However, such vision-based features are not specially designed for capturing properties in POMDPs/MDPs essential for decision making. To reveal structure that is particularly helpful for RL, many representation learning methods have been designed for different purposes, such as bi-simulation (Ferns et al., 2004; Gelada et al., 2019; Zhang et al., 2020), successor features (Dayan, 1993; Barreto et al., 2017; Kulkarni et al., 2016), spectral decomposition of transition operators (Mahadevan & Maggioni, 2007; Wu et al., 2018; Duan et al., 2019), latent future prediction (Schwarzer et al., 2020; Stooke et al., 2021) and contrastive learning (Oord et al., 2018; Mazoure et al., 2020; Nachum & Yang, 2021; Yang et al., 2021). These representation methods ignore the requirement of planning tractability. Moreover, they are learning based on a pre-collected dataset, which ignores the exploration issue.

Features that are able to represent value functions are desirable for efficient planning and exploration. Based on the linear MDPs structure (Jin et al., 2020b), several theoretical algorithms (Agarwal et al., 2020; Uehara et al., 2021) have been developed. Ren et al. (2022b); Zhang et al. (2022); Qiu et al. (2022); Ren et al. (2022a) bridge the gap between theory and practice and bypass computational intractability via different techniques, demonstrating advantages empirically. The proposed EPR is inspired from this class of representations, but extended to POMDPs, which is highly non-trivial.

## 5 EXPERIMENTS

Our experiments investigate how our algorithm performs in robotic lomocation simulation environments. We extensively evaluate the proposed approach on the Mojuco (Brockman et al., 2016) and DeepMind Control Suites (Tassa et al., 2018). We conduct experiments on both the fully observable MDP and partially observable POMDP settings.

### 5.1 FULLY OBSERVABLE MDP

**Dense-Reward Mujoco Tasks.** We first conduct experiments in the fully observable MDP setting in Mujoco locomotion tasks. This is a test suite commonly used for both model-free and model-based RL algorithms. We compare EPR with model-based RL baselines PETS (Chua et al., 2018) and ME-TRPO (Kurutach et al., 2018), and model-free RL baselines SAC (Haarnoja et al., 2018), TRPO (Schulman et al., 2015) and PPO (Schulman et al., 2017). In addition, we also compare to the representation learning RL baselines Deep Successor Feature (DeepSF) (Kulkarni et al., 2016) and SPEDE (Ren et al., 2022b). We list the best model-based RL results (except for iLQR (Li & Todorov, 2004)) in MBBL (Wang et al., 2019) for comparison. All algorithms are run for 200K environment steps. The results are averaged across four random seeds with a window size of 10K. We show that in Tab. 1, EPR significantly outperforms all the baselines including the strong previous SoTA model-free algorithm SAC.

Table 1: Performance on various MuJoCo control tasks. All the results are averaged across 4 random seeds and a window size of 10K. Results marked with $^*$ is adopted from MBBL. EPR achieves strong performance compared with baselines.

| | | HalfCheetah | Reacher | Humanoid-ET | Pendulum | I-Pendulum |
|---|---|---|---|---|---|---|
| Model-Based RL | ME-TRPO$^*$ | 2283.7±900.4 | -13.4±5.2 | 72.9±8.9 | **177.3±1.9** | -126.2±86.6 |
| | PETS-RS$^*$ | 966.9±471.6 | -40.1±6.9 | 109.6±102.6 | 167.9±35.8 | -12.1±25.1 |
| | PETS-CEM$^*$ | 2795.3±879.9 | -12.3±5.2 | 110.8±90.1 | 167.4±53.0 | -20.5±28.9 |
| | Best MBBL | 3639.0±1135.8 | **-4.1±0.1** | 1377.0±150.4 | **177.3±1.9** | **0.0±0.0** |
| Model-Free RL | PPO$^*$ | 17.2±84.4 | -17.2±0.9 | 451.4±39.1 | 163.4±8.0 | -40.8±21.0 |
| | TRPO$^*$ | -12.0±85.5 | -10.1±0.6 | 289.8±5.2 | 166.7±7.3 | -27.6±15.8 |
| | SAC$^*$ (3-layer) | 4000.7±202.1 | -6.4±0.5 | **1794.4±458.3** | 168.2±9.5 | -0.2±0.1 |
| Representation RL | DeepSF | 4180.4±113.8 | -16.8±3.6 | 168.6±5.1 | 168.6±5.1 | -0.2±0.3 |
| | SPEDE | 4210.3±92.6 | -7.2±1.1 | 886.9±95.2 | 169.5±0.6 | 0.0±0.0 |
| | **EPR** | **5107.6±195.4** | **-5.6±0.3** | 1494.6±131.3 | 169.4±4.1 | **0.0±0.0** |
| | | Ant-ET | Hopper-ET | S-Humanoid-ET | CartPole | Walker-ET |
| Model-Based RL | ME-TRPO$^*$ | 42.6±21.1 | 1272.5±500.9 | -154.9±534.3 | 160.1±69.1 | -1609.3±657.5 |
| | PETS-RS$^*$ | 130.0±148.1 | 205.8±36.5 | 320.7±182.2 | 195.0±28.0 | 312.5±493.4 |
| | PETS-CEM$^*$ | 81.6±145.8 | 129.3±36.0 | 355.1±157.1 | 195.5±3.0 | 260.2±536.9 |
| | Best MBBL | 275.4±309.1 | 1272.5±500.9 | **1084.3±77.0** | **200.0±0.0** | 312.5±493.4 |
| Model-Free RL | PPO$^*$ | 80.1±17.3 | 758.0±62.0 | 454.3±36.7 | 86.5±7.8 | 306.1±17.2 |
| | TRPO$^*$ | 116.8±47.3 | 237.4±33.5 | 281.3±10.9 | 47.3±15.7 | 229.5±27.1 |
| | SAC$^*$ (3-layer) | 2012.7±571.3 | 1815.5±655.1 | 834.6±313.1 | **199.4±0.4** | **2216.4±678.7** |
| Representation RL | DeepSF | 768.1±44.1 | 548.9±253.3 | 533.8±154.9 | 194.5±5.8 | 165.6±127.9 |
| | SPEDE | 806.2±60.2 | 732.2±263.9 | 986.4±154.7 | 138.2±39.5 | 501.6±204.0 |
| | **EPR** | **4081.3±973.9** | **2191.4±502.8** | **1326.3±20.8** | **200.8±0.1** | **1975.4±751.3** |

Table 2: Performance of on various Deepmind Suite Control tasks. All the results are averaged across four random seeds and a window size of 10K. Comparing with SAC, our method achieves even better performance on sparse-reward tasks.

| | | cheetah_run | cheetah_run_sparse | walker_run | walker_run_sparse | humanoid_run |
|---|---|---|---|---|---|---|
| Model-Based RL | Dreamer | 542.0 ± 27.7 | 499.9±73.3 | 337.7±67.2 | 95.4±54.7 | 1.0±0.2 |
| Model-Free RL | PPO | 227.7±57.9 | 5.4±10.8 | 51.6±1.5 | 0.0±0.0 | 1.1±0.0 |
| | SAC (2-layer) | 222.2±41.0 | 32.4±27.8 | 183.0±23.4 | 53.5±69.3 | 1.3±0.1 |
| | SAC (3-layer) | 595.2±96.0 | 419.5±73.3 | 700.9±36.6 | 311.5±361.4 | 1.2±0.1 |
| Representation RL | DeepSF | 295.3±43.5 | 0.0±0.0 | 27.9±2.2 | 0.1±0.1 | 0.9±0.1 |
| | Proto RL | 305.5±37.9 | 0.0±0.0 | 433.5±56.8 | 46.9±34.1 | 0.3±0.6 |
| | **EPR** | **611.6±53.5** | **469.8±30.6** | **792.8±35.7** | **701.8±30.4** | **11.5±5.4** |

In particular, we observe that most model-based algorithms have a hard time learning the walk and hop behavior in the Walker and Hopper environments respectively. We suspect that this is due to the fact that the quality of the data is bad at the initial data collection process (e.g., the agent often fall to the ground or has a hard time standing up). As a result, the behavior learned by most model-based algorithms can be suboptimal. For example, some model-based algorithms only learn to stand up without hopping in the Hopper environment. In contrast, EPR achieves SoTA performance in the Hopper task and Ant task, demonstrating the behavior of doing good exploration in the task domain.

**Sparse-Reward DM Control Tasks.** Manually-designed dense reward functions are extremely hard to obtain, while it is difficult to gain access to a good dense reward function in practical real-robot settings. Thus, exploration in the sparse-reward settings is a key consideration for the success of RL in robotics settings. We test our algorithm EPR with SAC and PPO in such cases. Here we compare with DeepSF as an additional representation RL baseline. Note that the critic network used in SAC and PPO is deeper than EPR. From Tab. 2, we see that EPR achieves a particularly huge gain compared to SAC and PPO in sparse reward tasks `walker-run-sparse`.

## 5.2 PARTIAL OBSERVABLE MDP COVERING VELOCITY

**Mujoco.** Often in practice, it is hard to recover a full observation of the states. Thus, the ability to handle a partially-observed MDP (POMDP) is also important if we can only recover partial observations. To conduct experiments in this setting, we mask the the velocities in the observations (replacing them by 0). We compare to algorithms with different embedding approaches that maps a given history sequence to a latent representation, which is used as the input for a SAC planner. We consider four embedding methods as baselines: Transformer (Trans), GRU, PSR (Guo et al., 2018), and finally a simple MLP baseline for sanity check, which concatenates the history sequence and directly maps that to a latent feature using a MLP. We find that this setting is very challenging and the

Table 3: Performance on various MuJoCo control tasks. All the results are averaged across 4 random seeds and a window size of 10K. Results marked with $^*$ is adopted from MBBL. EPR achieves strong performance compared with baselines.

| | | HalfCheetah | Humanoid-ET | Walker-ET |
|---|---|---|---|---|
| Representation RL | **EPR** | **3441.6 ± 993.0** | **865.6 ± 107.3** | 416.6 ± 145.6 |
| | PSR | 2679.75 ± 386 | 534.4 ± 36.6 | 862.4 ± 355.3 |
| Model-Free RL | MLP | 1612.0 ± 223 | 614.15 ± 67.6 | 236.5 ± 65.6 |
| | Trans | 1443.5 ± 227.2 | 387.1 ± 8.4 | 388.7 ± 224.9 |
| | GRU | 1664.3 ± 431.2 | 467.7 ± 43.4 | **1020.6 ± 364.9** |
| | | Ant-ET | Hopper-ET | S-Humanoid-ET |
| Representation RL | **EPR** | **1508.1 ± 594.4** | **1059.4 ± 582.5** | **805.7 ± 65.9** |
| | PSR | 1128.3 ± 166.6 | 818.8 ± 87.2 | 493.3 ± 65.2 |
| Model-Free RL | MLP | 1262.0 ± 68.7 | 260.6 ± 63.7 | 294.17 ± 49.4 |
| | Trans | 928.5 ± 44.2 | 470.8 ± 50.3 | 447.9 ± 112.6 |
| | GRU | 1190.8 ± 79.4 | 777.5 ± 113.3 | 485.9 ± 25.8 |

Table 4: Performance of on various Deepmind Suite Control tasks. All the results are averaged across four random seeds and a window size of 10K. Comparing with SAC, our method achieves even better performance on sparse-reward tasks.

| | | cheetah_run | cheetah_run_sparse | walker_run | walker_run_sparse | humanoid_run |
|---|---|---|---|---|---|---|
| Explicit Model | SLAC | 105.1 ± 30.1 | 0.0 ± 0.0 | 139.2 ± 3.4 | 0.0 ± 0.0 | 0.9 ± 0.1 |
| Model-Free RL | MLP | **743.3 ± 7.2** | 0.0 ± 0.0 | 279.8 ± 190.6 | 0.0 ± 0.0 | 1.2 ± 0.1 |
| | Trans | 379.6 ± 80 | 0.0 ± 0.0 | 68.06 ± 39.9 | 0.0 ± 0.0 | 0.92 ± 0.1 |
| Representation RL | PSR | 173.7 ± 25.7 | 0.0 ± 0.0 | 57.4 ± 7.4 | 0.0 ± 0.0 | 0.89 ± 0.1 |
| | **EPR** | 526.5±61.1 | **411.0±51.6** | **509.8±24.4** | **460.3±51.6** | **6.1±2.5** |

performance of all algorithms degrades compared to the fully-observable setting. Nevertheless, the proposed algorithm still achieves SoTA performance in tasks like Halfcheetah, Ant, SlimHumanoid. This demonstrates the capability of handling partial observability in EPR which can have an important effect in practice.

**DM Control Suite.** Correspondingly, we conduct POMDP experiments in the DM Control Suite. However, we find that covering all the velocities is very challenging and thus we cover only the last 3 dimensions of the velocity.

### 5.3 IMAGE-BASED ENVIRONMENTS

To test the capability of our method on image-based environments, we conduct an additional experiment on MetaWorld (Yu et al., 2020). We choose one of the `fetch-reach` tasks and compare against the model-free algorithm SAC+AE (Yarats et al., 2021) and a popular representation learning method SPR (Schwarzer et al., 2020). We show the results in Fig. 1 and note that the minimum distance between the current state and the goal is used as the evaluation metric (the smaller distance means better performance). We can see that EPR manages to reach the distant goal within 100K steps. Comparing to SAC+AE, EPR strictly dominate its performance. For SPR, although it learns faster at the beginning, EPR has better final performance.

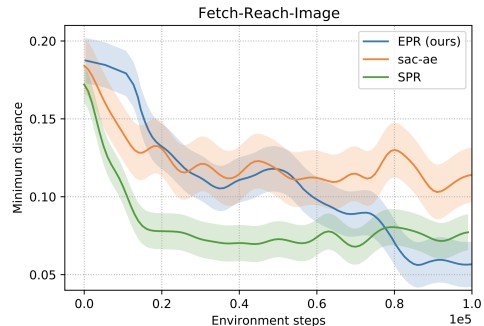

Figure 1: **EPR in image-based environment:** EPR gets a good performance compared to all baselines (e.g. SPR and SAC+AE).

### 6 CONCLUSION

We exploit Energy-based Predictive Representation (EPR) for linearly representing value functions for arbitrary policies and supporting reinforcement learning in partially observed environments with finite memories. The proposed EPR shows that planning and strategic exploration can be implemented efficiently. The coherent design of each component brings empirical advantages in RL benchmarks considering both the MDP and POMDP settings. Such superior performance makes the theoretical understanding of EPR more intriguing, which we leave as future work.

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

## A  MORE RELATED WORK

**Provably RL for POMDPs.**   Besides the statistical and computational hardness results for learning and planning upon POMDPs, most recent theoretical research focuses on overcoming the statistical complexity from the "curse of history" by considering tractable POMDPs (Krishnamurthy et al., 2016; Azizzadenesheli et al., 2016; Guo et al., 2016; Jin et al., 2020a; Liu et al., 2022). Similarly to the *observability* structure we exploited in our algorithm, these work bypass the curse of history by different special structures, reducing the whole history dependency to finite-length memory. Recently, Uehara et al. (2022); Wang et al. (2022) generalize these special structures with function approximation beyond tabular cases. Golowich et al. (2022a) consider the complexity planning and exploration together with learning, but only valid for tabular MDPs.

## B  DERIVATION OF THE RANDOM FEATURE IN EQUATION 8

We have that
$$P(o_{t+1}|x_t, a_t) = p(o_{t+1}) \exp\left(f(x_t, a_t)^\top \left(g(o_{t+1}) + \lambda f(x_t, a_t)\right)\right)$$
$$= p(o_{t+1}) \exp\left(\left(\lambda - \frac{1}{2}\right) \|f(x_t, a_t)\|^2\right) \exp\left(-\frac{\|g(o_{t+1})\|^2}{2}\right) \exp\left(\frac{\|f(x_t, a_t) + g(o_{t+1})\|^2}{2}\right),$$
where we have that
$$\exp\left(\frac{\|f(x_t, a_t) + g(o_{t+1})\|^2}{2}\right)$$
$$= (2\pi)^{-d/2} \exp\left(\frac{\|f(x_t, a_t) + g(o_{t+1})\|^2}{2}\right) \int \exp\left(-\frac{\|\omega - (f(x_t, a_t) + g(o_{t+1}))\|^2}{2}\right) d\omega$$
$$= (2\pi)^{-d/2} \int \exp\left(-\frac{\|\omega\|^2}{2} + \omega^\top (f(x_t, a_t) + g(o_{t+1}))\right) d\omega$$
$$= \mathbb{E}_{\omega \sim \mathcal{N}(0, I_d)} \left[\exp\left(\omega^\top f(x_t, a_t)\right) \exp\left(\omega^\top g(o_{t+1})\right)\right],$$
which concludes the proof for equation 8.

## C  OBSERVABLE LQG AS EPR

Follow the standard notations, the dynamics of Linear-Quadratic Gaussian is defined as
$$s_t = A s_{t-1} + B a_t + w_t,$$
$$o_t = C s_{t-1} + z_t,$$
where $w_t$ and $z_t$ are Gaussian noise. Define the matrix
$$G_L = [C^\top, CA^\top, \dots, \left(CA^{L-1}\right)^\top]^\top,$$
and reduced observation
$$\tilde{o}_t = o_t - z_t - C \left[\sum_{k=0}^{t-2} A^k B a_{t-k-1} + \sum_{k=0}^{t-2} A^k w_{t-k-2}\right].$$
By the observability condition of LQG, $G_L$ is full column rank, one can identify $s_0$ by
$$s_0 = \left(G_L^\top G_L\right)^{-1} \sum_{j=1}^{L} \left(A^\top\right)^{j-1} C^\top \tilde{o}_j.$$

Therefore, we have

$$s_1 = As_0 + Ba_0 + w_0 = A\left((G_L^\top G_L)^{-1} \sum_{j=1}^{L} (A^\top)^{j-1} C^\top \tilde{o}_j\right) + Ba_1 + w_0,$$

$$s_2 = As_1 + Ba_1 + w_1 = A^2\left((G_L^\top G_L)^{-1} \sum_{j=1}^{L} (A^\top)^{j-1} C^\top \tilde{o}_j\right) + ABa_1 + Ba_2 + Aw_0 + w_1,$$

$$s_{L+1} = As_L + Ba_L + w_L = A^L\left((G_L^\top G_L)^{-1} \sum_{j=1}^{L} (A^\top)^{j-1} C^\top \tilde{o}_j\right) + \sum_{j=0}^{L} A^{L-j} Ba_{j+1} + \sum_{j=0}^{L} A^{L-j} w_j,$$

$$o_{L+1} = Cs_{t+1} + z_t = CA^L\left((G_L^\top G_L)^{-1} \sum_{j=1}^{L} (A^\top)^{j-1} C^\top \tilde{o}_j\right) + C\sum_{j=0}^{L} A^{L-j} Ba_{j+1} + C\sum_{j=0}^{L} A^{L-j} w_j + z_t,$$

which means $o_{L+1}$ follows a Gaussian distribution with mean as a function of history $x_L = \left\{(o_{i-1}, a_i)_{i=1}^{L}\right\}$ and action $a_{L+1}$, and variance as a function of $\sigma_w$, $\sigma_z$, and $(A, B, C)$. Therefore, we have some function $f_{A,B,C,\sigma_w,\sigma_z}$ and $g_{A,B,C,\sigma_w,\sigma_z}$, such that

$$g_{A,B,C,\sigma_w,\sigma_z}(o_{L+1}) = f_{A,B,C,\sigma_w,\sigma_z}(x_L, a_{L+1}) + \xi, \quad \xi \sim \mathcal{N}(0, \mathbf{I}).$$

On the other hand, we set $\lambda = -\frac{1}{2}$, and $p(o) = \mathcal{N}(0, \mathbf{I})$ in (6), then, we obtain

$$p(o_{L+1}|x_L, a_L) \propto \exp\left(-\frac{\|g(o_{L+1}) - f(x_L, a_L)\|_2^2}{2}\right),$$

which reproduces the observable LQG with specific $f_{A,B,C,\sigma_w,\sigma_z}$ and $g_{A,B,C,\sigma_w,\sigma_z}$.

## D    EXPERIMENT DETAILS

### D.1    ONLINE SETTING

In Table 9, we list all the hyperparameters and network architecture we use for our experiments. We see that we don't use the additional exploration bonus term in the mojuco tasks. But this is very helpful in DM control suite tasks, especially in those sparse-reward tasks.

For evaluation in Mujoco, in each evaluation (every 5K steps) we test our algorithm for 10 episodes. We average the results over the last 4 evaluations and 4 random seeds. For Dreamer and Proto-RL, we change their network from CNN to 3-layer MLP and disable the image data augmentation part (since we test on the state space). The architecture we used for the transformer is following the Trajectory Transformer (Janner et al., 2021). The attention used is the causal attention. We tried to tune some of their hyperparameter (e.g., exploration steps in Proto-RL) and report the best number across our runs. However, due to the short time, it is also possible that we didn't tune the hyperparameter enough.

### D.2    LEARNING CURVES

We provide the performance curves for online DM Control Suite experiments in Figure 2. As we can see in the figures, the proposed EPR converges faster and achieve the state-of-the-art performances in most of the environments, demonstrating the sample efficiency and the ability to balance of exploration vs. exploitation of EPR. We also provide additional curves for POMDP setting in Figure 3.

### D.3    IMAGE-BASED EXPERIMENTS

We provide the details of metaworld image-based experiments here. We first provide an illustration of the reach environment in Figure 4. We then provide some more experiment details in the following section.

Table 5: Hyperparameters used for EPR in all the environments in MuJoCo and DM Control Suite.

|  | Hyperparameter Value |
| --- | --- |
| Bonus Coefficient (MuJoCo) | 0.0 |
| Bonus Coefficient (DM Control) | 5.0 |
| Actor lr | 0.0003 |
| Model lr | 0.0003 |
| Actor Network Size (MuJoCo) | (256, 256) |
| Actor Network Size (DM Control) | (1024, 1024) |
| ERP Embedding Network Size (MuJoCo) | (1024, 1024, 1024) |
| ERP Embedding Network Size (DM Control) | (1024, 1024, 1024) |
| Critic Network Size (MuJoCo) | (1024, 1) |
| Critic Network Size (DM Control) | (1024, 1) |
| Discount | 0.99 |
| Target Update Tau | 0.005 |
| Model Update Tau | 0.005 |
| Batch Size | 256 |

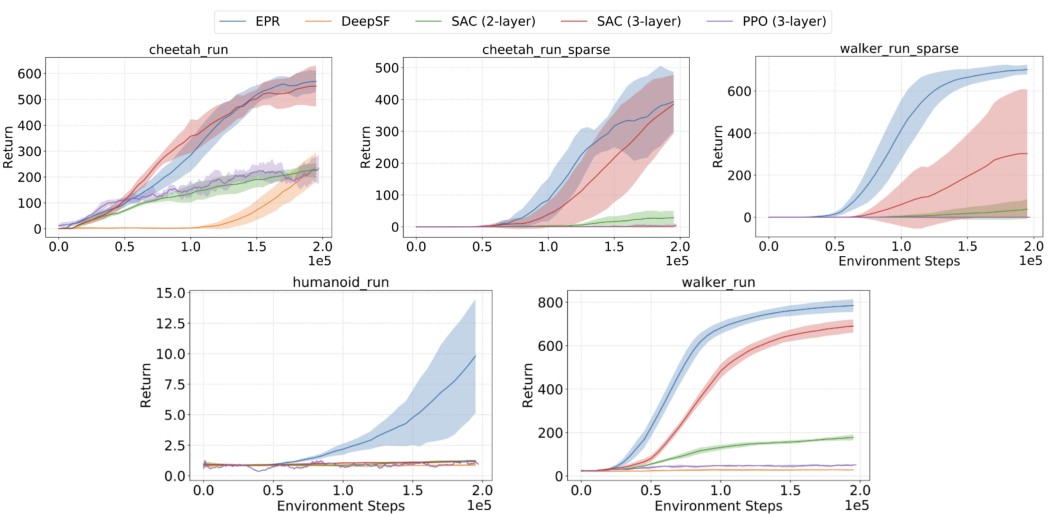

Figure 2: Performance Curves for online DM Control Suite.

Table 6: Settings of adapted OpenAI Fetch-Reach Environment.

|  | Hyperparameter Value |
| --- | --- |
| Maximum Episode Steps | 50 |
| Reward Type | 'sparse' |
| Observation Size | (3, 64, 64) |
| Fixed Goal Position | (1.27, 0.90, 0.66) |

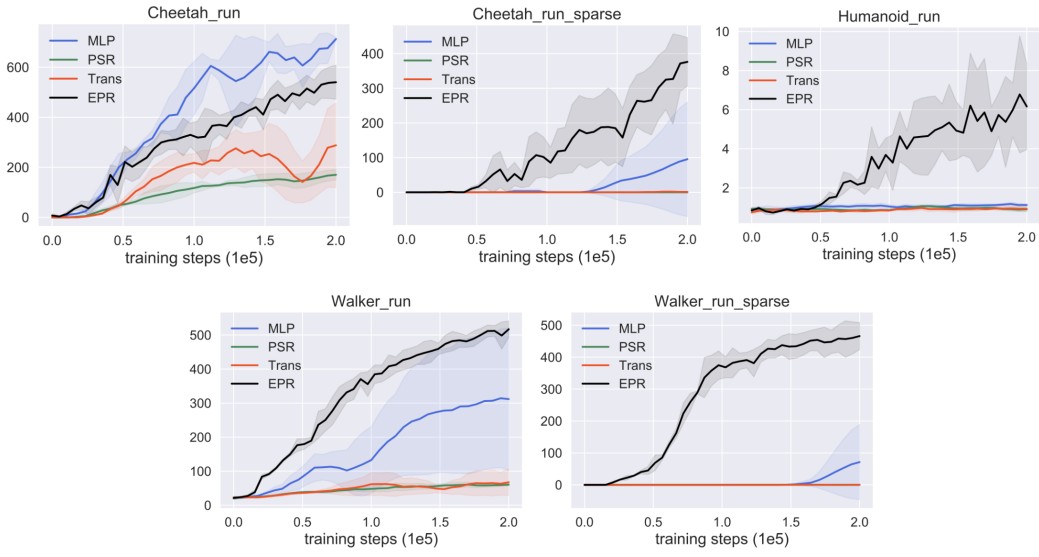

Figure 3: Performance Curves for online POMDP DM Control Suite.

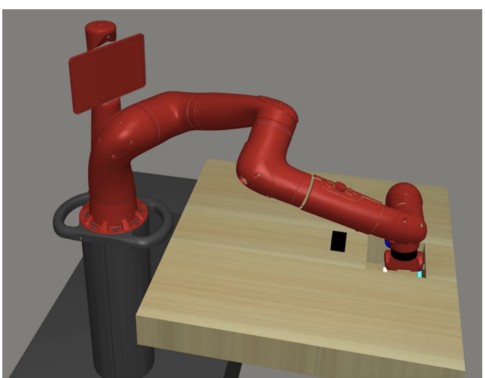

Figure 4: **Reach environment:** Using a robot arm to reach a specific position.

Table 7: Hyperparameters used for EPR in FetchReachImage.

|  | Hyperparameter Value |
| --- | --- |
| Bonus Coefficient (MuJoCo) | 0.0 |
| Bonus Coefficient (DM Control) | 5.0 |
| Actor lr | 0.0003 |
| Model lr | 0.0003 |
| Actor Network Size (MuJoCo) | (256, 256) |
| Actor Network Size (DM Control) | (1024, 1024) |
| ERP Embedding Network Size (MuJoCo) | (1024, 1024, 1024) |
| ERP Embedding Network Size (DM Control) | (1024, 1024, 1024) |
| Critic Network Size (MuJoCo) | (1024, 1) |
| Critic Network Size (DM Control) | (1024, 1) |
| Discount | 0.99 |
| Target Update Tau | 0.005 |
| Model Update Tau | 0.005 |
| Batch Size | 256 |

Table 8: Hyperparameters used for SPR in FetchReachImage.

|  | Hyperparameter Value |
| --- | --- |
| lr | 0.0001 |
| Dropout | 0.5 |
| Discount | 0.99 |
| Batch Size | 32 |
| Augmentation | off |
| Target Update Tau | 0.005 |
| Model Update Tau | 0.005 |
| Batch Size | 256 |
| Update | Distributional Q |
| Dueling | True |
| Optimizer | Adam |
| Optimizer: learning rate | 0.0001 |
| Max gradient norm | 10 |
| Priority exponent | 0.5 |
| Noisy nets parameter | 0.5 |
| Min replay size for sampling | 2000 |
| Replay period every | 1 step |
| Updates per step | 2 |
| Multi-step return length | 10 |
| Q network: channels | 32, 64, 64 |
| Q network: filter size | $8 \times 8, 4 \times 4, 3 \times 3$ |
| Q network: stride | 4, 2, 1 |
| Q network: hidden units | 256 |
| Non-linearity | ReLU |
| Target network: update period | 1 |
| $\lambda$ (SPR loss coefficient | 2 |
| K (Prediction Depth) | 5 |

Table 9: Hyperparameters used for SAC-AE in FetchReachImage.

|  | Hyperparameter Value |
| --- | --- |
| Critic lr | 0.001 |
| Actor lr | 0.001 |
| Discount | 0.99 |
| Batch Size | 128 |
| Critic Q-function soft-update rate $\tau_Q$ | 0.01 |
| Critic encoder soft-update rate $\tau_{enc}$ | 0.05 |
| Critic target update frequency | 2 |
| Actor update frequency | 2 |
| Actor standard deviation bounds | $[-10, 2]$ |
| Autoencoder learning rate | 0.001 |
| Temperature learning rate | 0.0001 |
| Temperature Adam's $\beta_1$ | 0.5 |
| Init temperature | 0.1 |

