# OpenReview forum: "Energy-based Predictive Representation for Reinforcement Learning"
_ICLR.cc/2023/Conference — Submitted to ICLR 2023_

### Official Review · Reviewer_m9BZ · 2022-10-15

**Confidence:** 4
**Correctness:** 1
**Technical Novelty And Significance:** 2
**Empirical Novelty And Significance:** 2
**Recommendation:** 3

**Clarity, Quality, Novelty And Reproducibility:**

Equation 6 to 18 are not explained clearly. Among others, here are some elements that are not clear:
- Right after Equation 6, it is written that "p(o) is a fixed distribution". If it's fixed, can you give it more precisely? In addition, right after equation 18, it is written that "$p(o)$ is a mixture of replay buffer and collected trajectories". (minor: what does it mean i=1 in $\\{o_i\\}^{m}_{i=1} \sim p(o)$?)
- What are $f$ and $g$ in Equation  6 and 7? In Algorithm 1, they have an additional subscript $\theta$. If these mean that $\theta$ are parameters of the function approximators $f$ and $g$, I assume that it shouldn't be the same parameters?
- What is $\psi_\omega$?

$\gamma-observability$ is defined in Equation 1 but is not used anywhere in the paper?

What is a "SAC planner"? SAC is a model-free algorithm as far as I'm aware of.

Minor comments:
- The notation $b(x_t)$ was never used before
- In the following "Initially, given a state $s_1 \sim \mu(s)$ as a starting point, (...)", should it be $s_0 \sim \mu(s)$ since you define $h ∈ [0, H ]$?
- Assumption 1: what is $b$, what is $b'$? It looks like the state space (for the belief) needs to the same dimension than the observation space?
- line 3 in Algorithm 1: Shouldn't there be rewards also collected in the dataset?
- Figure 1: How many seeds?

Typos:
- 1st sentence: "Reinforcement learning (RL) based on Markov Decision Processes (MDPs) has has proved to be"
- "W propose Energy-based Predictive Representation (EPR)" leverages linear structure in the dynamics
- "onecan write"

**Strength And Weaknesses:**

Strengths:
- The claims are very ambitious

Weaknesses:
- The paper does not provide a clear formalization and description of the algorithm

**Summary Of The Paper:**

The paper aims at using "Energy-based Predictive Representation (EPR)" to create a "unified approach to practical reinforcement learning algorithm design for both the MDP and POMDP settings, which enables learning, exploration, and planning to be handled in a coherent way".

**Summary Of The Review:**

The paper is not clear.

---

> ### Author Response · Authors · 2022-11-19
> **Author Response**
>
> We thank the reviewer for the feedback.
>
> **The paper does not provide a clear formalization and description of the algorithm. ** The algorithm derivation is listed in section Section 3. We have also provided the algorithm box in Algorithm 1. We also add more implementation details in Appendix D.
>
> **Clarification on Eq.6 and Eq. 18.** p(o) is the distribution for sampling in MLE in Eq. 17. Since we only need samples, in practice, we treat the replay buffer as $p(o)$ in our implementation.
>
> ${o_i}_{i=1}^m \sim p(o)$ means we sample $o_i, i=1,...m$, from $p(o)$
>
> **What are f and g in Equation 6 and 7?** The f and g are separate neural networks in Equations 6 and 7. The $\theta$ denotes the parameters of the neural network. We are sorry for the confusion and we have revised the draft for clarification in Algorithm 1.
>
> **What is $\psi_w$?** We defined $\psi_w$ in equation 10.
>
> **$\gamma$−observability is defined in Equation 1 but is not used anywhere in the paper?** The gamma-observability is introduced in preliminary for observalibity pomdp in [Golowich et al. 2022]. We exploit the \gamma-observability structure to justify that the finite-memory truncation approximation is reasonable for POMDP in Section 2 below Assumption 1.
>
> **What is a "SAC planner"?** We believe this is a terminology issue. Here we refer to “planner” as the algorithm for finding the optimal policy upon the MDP following the definition in RL community (https://rltheory.github.io/lecture-notes/planning-in-mdps/lec1/#:~:text=Computing%20an%20optimal%20policy%20can,the%20algorithmic%20question%20becomes%20interesting!), which is what the SAC algorithm is doing.
>
> **what is b, what is b$\prime$?** b and b$\prime$ are two distributions over states. The Assumption 1 is introduced in the existing literature [Golowich et al, 2022] to characterize some structure property of a class of POMDP, in which finite-memory truncation is reasonable.
>
> **Figure 1: How many seeds?** We averaged across 4 seeds.
>
> **Typos** We have revised the paper accordingly.

---

### Official Review · Reviewer_TDvH · 2022-10-27

**Confidence:** 3
**Correctness:** 3
**Technical Novelty And Significance:** 3
**Empirical Novelty And Significance:** 3
**Recommendation:** 6

**Clarity, Quality, Novelty And Reproducibility:**

The paper is overall clear, but I think more intuition about the type of environment/tasks where EPR does or does not work is needed. Given the that architectures used to estimate parameters are not complicated, I do not see a major reproducibility issue. In terms of novelty, I think making a method that works well for Partially Observable environments is extremely valuable. However, I think the experiments should be more focused on this part as the MDP part is not new.

**Strength And Weaknesses:**

Strength:
The ideas behind taking advantage of the environmental structure were very interesting. Moreover, deep networks are still not performing well in domains with uncertainty, making this line of work very worthy of more exploration. Finally, I think the authors did a very good job of explaining the connection between EPR and other existing representations (although these connections could be moved to the appendix to make room for explaining the domains that EPR works/does not work)

Weakness:
I think the paper could be significantly improved if the authors give more intuition about the type of tasks in which their method works and tasks where it does not, especially classic POMDP tasks such as RockSample or classic localization problems. Particularly, it looks like the belief should be concentrated around a state and its surrounding (as opposed to multi-modal), so the method works well.

I also found the choice of masking velocity to make the tasks POMDP in experiments a little bit strange. What was the rationale behind that? Generally, while I do believe that taking advantage of real problem structures is totally okay and in fact useful, uncertainty about agents' own sensing is not very much aligned with reality. There is actually a very nice benchmark for POMDPs based on MuJoCo tasks in "Recurrent Model-Free RL Can Be a Strong Baseline for Many POMDPs" by Ni et al, ICML 2022 (repo: https://github.com/twni2016/pomdp-baselines). They have defined different types of Partial Observabilty and I think using even one type of these test-bench marks is extremely valuable for evaluation. Also, it looks like some baseline methods are provided by them.

**Summary Of The Paper:**

The paper offers a method for reinforcement learning under partial observability by offering a new form of representing the environment model. Specifically, this representation takes advantage of the structure of the environment and linear approximation of value/Q functions. The authors showed the superiority of their method on some MuJoCo tasks.

**Summary Of The Review:**

Overall, I believe this is a very interesting and promising idea, but I wish to see more standard experiments and/or more intuition about the type of tasks/environments where EPR is useful.

---

> ### Author Response · Authors · 2022-11-19
> **Author Response**
>
> We thank the reviewer for the feedback.
>
> **I think the paper could be significantly improved if the authors give more intuition about the type of tasks in which their method works and tasks where it does not.** Belief is the posterior distribution of states given current observations, which may not be concentrated. To bypass the difficulty in representing the belief explicitly (infinite-dimension and multi-modal), we exploit the effective representation in terms of value function. The major benefit of our method comes from this reformulation for representation, learning, planning and exploration.
>
> **I also found the choice of masking velocity to make the tasks POMDP in experiments a little bit strange. What was the rationale behind that?**
> Masking velocity is a common setting when constructing PODMP environments [1, 2, 3]. We choose to mask the velocity Mujoco and DM Control, which is more aligned with practical settings. In fact, for practical usage, measuring the velocity of the robot can be harder than measuring the position.
> We thank the reviewer for bringing [1] up. We have compared with recurrent RL as our baseline in Table. 3, Section 5.2. We choose GRU as the implementation baseline. We also tried LSTM but found its performance is slightly worse than GRU.
>
> [1] Ni, Tianwei, Benjamin Eysenbach, and Ruslan Salakhutdinov. "Recurrent model-free rl can be a strong baseline for many pomdps." International Conference on Machine Learning. PMLR, 2022.
>
> [2] Gangwani, Tanmay, et al. "Learning belief representations for imitation learning in pomdps." Uncertainty in Artificial Intelligence. PMLR, 2020.
>
> [3] Weigand, Stephan, et al. "Reinforcement Learning using Guided Observability." arXiv preprint arXiv:2104.10986 (2021).

---

### Official Review · Reviewer_XFFG · 2022-11-01

**Confidence:** 4
**Correctness:** 3
**Technical Novelty And Significance:** 3
**Empirical Novelty And Significance:** 2
**Recommendation:** 6

**Clarity, Quality, Novelty And Reproducibility:**

- The paper is not hard to follow and provides numerous citations to the (many) prior works that it builds upon.
- The paper seems technically sound. It details with precision the prior results is makes use of, and new derivations are included in the text. One issue is the typos/language, which can nevertheless be fixed with careful proofreading (and some other minor errors in weaknesses listed above).
- The work is quite novel. The idea of spectral factorization with linear features is not new, but this paper explores this approach in the new context of POMDPs. Furthermore, the overall framework is not a straightforward application of existing ideas but there are novel elements involved, such as the derivation of a linear factorization for the transition function for the beliefs.
- Empirical results are strong and the comparison with baselines quite extensive.
- The authors provide the experimental settings/hyperparameters that are necessary for replicating their results.

**Strength And Weaknesses:**

Strengths
- The proposed framework builds upon several prior works, e.g., prior work on linear/low-rank structures and spectral factorization for MDPs, random Fourier features, elliptical confidence bounds, feature space augmentation. The authors explain the various steps for building their framework in detail, and provide plenty of references in the process. This makes the paper not hard to follow.
- Related work from several domains is covered quite thoroughly. Connections to prior works on representation learning (e.g., PSR and SPEDE) are also established.
- The proposed framework seems able to solve a real problem for POMDPs, especially as the size of the state space becomes large. Furthermore, the 2 assumptions of \gamma-observability and linearity are meaningful and well motivated by prior work.
- The empirical results show strong performance. I particularly like that the EPR framework can be beneficial not just for POMDPs, but for fully observable MDPs as well.

Weaknesses
- There are quite a few typos. I suggest the authors proofread the document carefully. Examples: Th predictive -> The predictive, has has proved -> has proved, with slightly abuse -> with slight abuse, W propose -> We propose, an straightforward extension -> a straightforward extension, etc.
- The notation was confusing at times. For instance, the authors introduced a symbol x_t^L for fixed-window histories (of size L) but then used x_t in equations (6) and (7) (and even elsewhere). I think consistency in the notation everywhere in the paper would help.
- I think there is an (easy to fix) error in Assumption 1. \gamma-observability needs a \gamma multiplicative factor in the right term of the inequality. This is currently missing.

**Summary Of The Paper:**

The paper proposes Energy-based Predictive Representation (EPR) for tractable learning, planning and exploration in POMDPs and fully observable MDPs. Concretely, the authors propose a nonlinear energy-based learnable model which does not explicitly parameterize the beliefs. Their model provides a linear sufficient representation for the state-action value function, which can be particularly beneficial to large state spaces. Subsequently, the authors introduce computationally tractable algorithms for learning, exploration/exploitation and planning, based on the principles of optimism and pessimism in the face of uncertainty for online and offline RL. The paper also discusses the connections between EPR and Predictive State Representation (PSR), and points out that the latter is not generally associated with efficient exploration and planning algorithms. The authors conduct an extensive empirical evaluation with both fully observable MDPs and POMDPs, comparing their approach to several model-based, model-free and representation-learning frameworks. The results confirm the superior empirical performance of the proposed representation learning scheme.

**Summary Of The Review:**

Overall, I believe that the current work is a very interesting addition in the literature of representation learning for POMDPs with large state spaces, with strong and promising empirical results.

---

> ### Author Response · Authors · 2022-11-19
> **Author Response**
>
> We thank the reviewer for the feedback. We have revised the paper accordingly.

---

### Official Review · Reviewer_4vmU · 2022-11-03

**Confidence:** 4
**Correctness:** 2
**Technical Novelty And Significance:** 2
**Empirical Novelty And Significance:** 2
**Recommendation:** 3

**Clarity, Quality, Novelty And Reproducibility:**

Clarity and Quality can be improved by properly dealing with the comments above.

My concern regarding novelty is that without detailed motivation and explanation, the proposed method might be seen as a combination of an existing energy-based model and random feature techniques.

Reproducibility can be improved by explicitly writing the source of the baseline code used and submitting the implementation code as supplementary material.

**Strength And Weaknesses:**

The authors dealt with the formulation of RL in POMDPs with various interesting concepts. However, several parts of the paper should be clarified, and additional experiments should be performed. It was hard to check the reproducibility. Please refer to the detailed comments below.


Comments

1. The core idea of linear value-function representation is not well-motivated. In this paper, the following statement is written: "To handle the complexity induced by large state spaces, linear/low-rank structures have been introduced in MDPs." However, several questions arise.
- Is linear representation guaranteed to be effective in POMDPs? Since linear representation lacks representation ability compared with nonlinear counterparts, it can be inefficient in POMDPs, and even in MDPs with large spaces.
- Does the proposed linear representation enjoy better Q-function convergence?
- This statement contradicts the core idea of linear representation. “In practice, we perform fitted Q iteration with a nonlinear component extending the linear parameterization.” (page 6)

2. Regarding energy-based parameterization
- “We avoid any explicit parameterization and computation of beliefs” (page 4). To support the superiority of the energy-based model over explicit parameterization, we require explicit parameterization methods including e.g., DVRL (Igl et al., 2018) or SLAC (Lee et al., 2020).
- The background of energy-based parameterization should be explained in detail (e.g., in Preliminaries) since it is one of the core components in the proposed method.

3. Regarding the random Fourier feature trick
- It would be better to explicitly write the full derivation of Eqs. 9 and 10 for the readers since this is also another core component of the proposed method.
- What is the sample number $n$ in the random Fourier feature trick? Is the performance robust to  $n$?

4. Regarding exploration
- Is extracting the random feature $\phi$ the best method for generating additional bonus reward? Any other intrinsic reward method should be compared to support the claim.

5. Experiments
- Instead of the choice of masking velocity to make the tasks POMDP in experiments, is the proposed method still effective when masking position$^{a}$ (and let velocity visible), or input with noise$^{b}$?

a. Han et al., Variational Recurrent Models for Solving Partially Observable Control Tasks

b. Meng et al., Memory-based Deep Reinforcement Learning for POMDP

- As stated above, other explicit belief parameterization methods and other intrinsic reward for exploration are required.
- In Tables 3 and 4, please describe the details of MLP and Trans. Does the transformer use causal encoding? Is positional encoding used? What is the number of layers and multi-heads of Trans?
- Why the proposed method is effective in MDPs? Is that because of linear representation?
- What is the number of negative samples $m$? Is the performance robust to  $m$ (and $\lambda, \alpha$)?

6. Notation
- Is $x_t$ implemented using RNN?
- $\gamma$ is missing in Assumption 1.


**Summary Of The Paper:**

This paper proposes a new algorithm dealing with reinforcement learning (RL) in partially observable environments by linearly representing action-value functions. To learn linear action-value functions, the energy-based model is applied to represent a valid transition function. This energy-based method bypasses the explicit computation of beliefs. Then the learned energy-based transition on beliefs can be linearly factorized by applying the existing random Fourier feature trick. The action-value function and additional reward are represented by using the random feature, and numerical results show promising results on exploration, exploitation, and planning in RL with MDPs/POMDPs.

**Summary Of The Review:**

I have several concerns about the core idea as well as clarity, novelty, and reproducibility, and I hope the authors can address this in their rebuttal.

---

> ### Author Response · Authors · 2022-11-19
> **Author Response**
>
> We thank the reviewer for the feedback.
>
> **Novelty**
> We respectfully disagree with the claim about novelty. As we discussed in the paper, the energy-based representation is proposed for representing state-value function with computational-efficient learning, exploration and planning for POMDP, which has never been proposed in literature.  Moreover, it generalized the existing work, e.g., SPEDE (Ren et. al, 2022) for general noise models beyond Gaussian and partial observation cases, as one of our major contributions. It is unfair to simply recognize the proposed method as  “a combination of an existing energy-based model and random feature techniques”, as such combination is following deep mathematical derivation and has never been exploited in literature.
>
> **The flexibility of linear representation for POMDP.**
> We first emphasize this claim is incorrect. In fact, for EBMs we discussed, it is highly nonlinear and flexible, and we can find the feature to linearly represent the Q-function. The “linear” here is to emphasize it is the linear space for Q function. However, representation with respect to the input can be nonlinear, which guarantees that it can have sufficient representation power for the Q function.
>
> **Does the proposed linear representation enjoy better Q-function convergence?**
> As we iterate in the main text,  the Q-function can be linearly represented. Therefore,  the dynamic programming will be better converge to the target, as now we only need to perform a relatively simple regression on top of the representation if the representation are learned properly.
>
> **This statement contradicts the core idea of linear representation.**
> As we explained in the main text, for arbitrary policy, we can exploit the learned feature to represent its corresponding Q-function *linearly*, which is the reason to name the learned feature linear representation.
>
> With the exploration bonus, theoretically, we can still represent the bonus augmented Q function with a reward augmented feature. We only follow CTRL(Zhang et al. ICML 2022) for computation efficiency.
>
> **We require explicit parameterization methods including e.g., DVRL (Igl et al., 2018) or SLAC (Lee et al., 2020).**
> We ran an additional experiment of SLAC on the POMDP DM Control tasks. We didn’t run DVRL since the performance of SLAC is always better than DVRL in the SLAC paper (Figure. 3 in the paper). We found that the performance of SLAC is strictly worse than our method. The new results have been updated in Table. 4. We also paste the results here for reference.
> |   |       | cheetah\_run             | cheetah\_run\_sparse    | walker\_run             | walker\_run\_sparse     | humanoid\_run        |
> |---|-------|--------------------------|-------------------------|-------------------------|-------------------------|----------------------|
> |   | SLAC  | 105.1 $\pm$ 30.1         | 0.0 $\pm$ 0.0           | 139.2 $\pm$ 3.4         | 0.0 $\pm$ 0.0           | 0.9 $\pm$ 0.1        |
> |   | MLP   | **743.3 $\pm$ 7.2** | 0.0 $\pm$ 0.0           | 279.8 $\pm$ 190.6       | 0.0 $\pm$ 0.0           | 1.2 $\pm$ 0.1        |
> |   | Trans | 379.6 $\pm$ 80           | 0.0 $\pm$ 0.0           | 68.06 $\pm$ 39.9        | 0.0 $\pm$ 0.0           | 0.92 $\pm$ 0.1       |
> |   | PSR   | 173.7 $\pm$ 25.7         | 0.0 $\pm$ 0.0           | 57.4 $\pm$ 7.4          | 0.0 $\pm$ 0.0           | 0.89 $\pm$ 0.1       |
> |   | EPR   | 526.5$\pm$61.1           | **411.0$\pm$51.6** | **509.8$\pm$24.4** | **460.3$\pm$51.6** | **6.1$\pm$2.5** |
>
> **The background of energy-based parameterization should be explained in detail.**
> We have updated the paper accordingly in the preliminaries section.
>
> **It would be better to explicitly write the full derivation of Eqs. 9 and 10.**
> We have updated the paper accordingly with details in Appendix B.
>
> **What is the sample number n in the random Fourier feature trick? Is the performance robust to N?**
> There is a tradeoff between the performance vs. memory cost w.r.t. N. Specifically, we use a random feature of 1024 in our experiments. We found the performance saturates when N is large enough (e.g., 1024). We’ll add the ablation with respect to N in our next revision.
>
> **Include other intrinsic reward methods.**
> We included SAC with entropy based exploration. (We have added the comparison with RND and ICM, which are heuristically designed for state-based exploration, while the proposed UCB bonus is more effective within the learned feature space). However, we would like to emphasize this is an orthogonal issue, as the purpose of the paper is to learn a *representation* for better planning and exploration, while the other existing representations of POMDP does not take this into account.

---

> ### Author Response · Authors · 2022-11-19
> **Author Response**
>
> **Is the proposed method still effective when masking position (and let velocity visible), or input with noise?**
> The proposed algorithm is independent w.r.t. the missing information, and can be applied for arbitrary masking.
>
> Masking velocity is a common setting when constructing PODMP environments [1, 2, 3]. We choose to mask the velocity Mujoco and DM Control, which is more aligned with practical settings, where the velocity can not be directly obtained from observations, in contrast to the position.
>
> [1] Ni, Tianwei, Benjamin Eysenbach, and Ruslan Salakhutdinov. "Recurrent model-free rl can be a strong baseline for many pomdps." International Conference on Machine Learning. PMLR, 2022.
>
> [2] Gangwani, Tanmay, et al. "Learning belief representations for imitation learning in pomdps." Uncertainty in Artificial Intelligence. PMLR, 2020.
>
> [3] Weigand, Stephan, et al. "Reinforcement Learning using Guided Observability." arXiv preprint arXiv:2104.10986 (2021).
>
> **In Tables 3 and 4, please describe the details of MLP and Trans. Does the transformer use causal encoding? Is positional encoding used? What is the number of layers and multi-heads of Trans?**
> The MLP baseline concatenates the history sequence and directly maps that to a latent feature using a MLP.
>
> The Transformer baseline uses causal encoding. The number of layers and number of heads are one. Positional encoding is used. These details can be found in Appendix. D1.
>
> **Why the proposed method is effective in MDPs? Is that because of linear representation?**
> As we discussed in the main text in page 5-6, the proposed method will be degraded to linear representation for linear MDP, which is more general than SPEDE (Ren et al., 2022). In this case, the Q-function can be represented linearly for arbitrary stochastic dynamics, and thus, more efficient in planning and exploration.
>
> We would like to re-emphasize that learning linear representation in linear MDP is not resulting in a linear model, but an nonlinear EBM, which is highly flexible.
>
> **What is the number of negative samples m? Is the performance robust to m (and $\lambda$, $\alpha$)?**
> The negative number m is 512. In our experiments, the performance is pretty robust to m and also ($\lambda$, $\alpha$). We’ll update the paper in our next revision with these ablations.
>
> **Is xt implemented using RNN?**
> Yes, $X_t$ is implemented using RNN (e.g., GRU).
>
> **$\gamma$ is missing in Assumption 1.**
> We have revised the paper.
>
> **Reproducibility** We have uploaded a zip for in the supplementary materials containing the code.

---

### Decision · Program_Chairs · 2023-01-20

**Decision:**

Reject

**Justification For Why Not Higher Score:**

The paper combines a number of existing techniques that make it hard to follow and tell the novelty. The authors' responses did not fully resolve the reviewer's concerns. The uploaded zip file is empty (instead of containing code).

**Justification For Why Not Lower Score:**

N/A

**Metareview: Summary, Strengths And Weaknesses:**

Summary:
This paper proposes Energy-based Predictive Representation (EPR) to for RL in POMDPs to be tractable. The model bypasses the explicit computation of beliefs and the learned energy-based transition on beliefs can be linearly factorized by applying the existing random Fourier feature trick. Then optimism and pessimism are applied to learn online or offline RL. The authors conduct an extensive empirical evaluation with both fully observable MDPs and POMDPs, comparing their approach to several model-based, model-free, and representation-learning frameworks. The results confirm the superior empirical performance of the proposed representation learning scheme.

Strength:
- The proposed framework seems able to solve a real problem for POMDPs.
- The empirical results show strong performance.

Weakness:
- The core idea of linear value-function representation is not well-motivated.
-  typos.
- Some of the notations were confusing.

**Summary Of Ac-Reviewer Meeting:**

Reviewers have mixed ratings of the paper, but a consensus was reached -- authors' rebuttal address some reviewers' comments but not all the critical ones.